

# Free fermions at the edge of interacting systems

**Jean-Marie Stéphan**[1][*]

**1** Univ Lyon, CNRS, Université Claude Bernard Lyon 1,
UMR5208, Institut Camille Jordan, F-69622 Villeurbanne, France

[*] stephan@math.univ-lyon1.fr

## Abstract

We study the edge behavior of inhomogeneous one-dimensional quantum systems, such as Lieb-Liniger models in traps or spin chains in spatially varying fields. For free systems these fall into several universality classes, the most generic one being governed by the Tracy-Widom distribution. We investigate in this paper the effect of interactions. Using semiclassical arguments, we show that since the density vanishes to leading order, the strong interactions in the bulk are renormalized to zero at the edge, which simply explains the survival of Tracy-Widom scaling in general. For integrable systems, it is possible to push this argument further, and determine exactly the remaining length scale which controls the variance of the edge distribution. This analytical prediction is checked numerically, with excellent agreement. We also study numerically the edge scaling at fronts generated by quantum quenches, which provide new universality classes awaiting theoretical explanation.



# 1   Introduction

The celebrated Tracy-Widom (T-W) distribution [1] was originally discovered while studying the largest eigenvalue of large random matrices. More precisely, it describes in this context the appropriately rescaled cumulative distribution function of the largest eigenvalue $\lambda_N^{\max}$ of a random $N$ by $N$ gaussian hermitian matrix, in the limit $N \to \infty$:

$$E(s) = \lim_{N \to \infty} \text{Proba}\left( \frac{\lambda_N^{\max} - \sqrt{2N}}{2^{-1/2} N^{-1/6}} \leq s \right). \tag{1}$$

The appearance of this distribution is not at all limited to random matrix theory. In fact, such a universal scaling occurs in edge problems as diverse as increasing subsequences of random permutations [2], growth models [3–5], dimer coverings on graphs [6], classical exclusion processes [7], or quantum quenches [8,9], to name a few. In those problems, the T-W distribution describes the edge properties of a macroscopic $2d$ classical system at equilibrium, or the front of a $1d$ system out of equilibrium. From a mathematical perspective T-W is based on a determinantal point process (free fermions in physicist parlance), with a correlation kernel (propagator) known as the Airy kernel. While the diversity of problems where this distribution appears looks impressive, most of those are free fermions in disguise. A simple physical picture was put forward in [3,10] (see also [11] for an earlier related work). In such a picture the Airy kernel naturally emerges as a "filter" that projects onto the negative energy eigenstates of a free fermion model in a linear potential. Showing convergence to T-W in those free problems then boils down to showing convergence of the correlation kernel to the Airy kernel, after appropriate edge rescaling.

The aim of the present paper is to investigate several examples of physical $1d$ *interacting* quantum mechanical models where the T-W distribution naturally appears in the ground state. This will be done by combining heuristic semiclassical and thermodynamic Bethe Ansatz arguments, supplemented by careful numerical checks. The main reason why this is possible follows from a simple –but difficult to prove– renormalization argument: particles, say in a trap, are typically diluted near the edge, so are less sensitive to the effects of interactions which might be otherwise very strong in the bulk. We will also investigate what happens when those interacting quantum systems are put out of equilibrium, which can lead to more complicated and much less understood universality classes.

This long introduction is devoted to the free case, which helps put all the important concepts in place –once this is done treating interacting systems will prove no more complicated, since the edge will turn out to be free in the end. It is organized as follows. In section 1.1 we introduce the free fermion model which has the Airy kernel as correlation kernel. We then present a derivation of the exact Fredholm determinant formula for the Tracy-Widom distribution (section 1.2), and briefly discuss various extensions. Finally, we explain on a simple

example how T-W scaling occurs at the edge of a realistic fermion model (section 1.3). The mechanism for this is more important than the specific derivation, and follows from general semiclassical arguments. Let us stress that this introduction does not contain new results and follows Ref. [10] to some extent; the only slight originality lies in the use of the language of field theory and Wick's theorem.

## 1.1 Free Airy fermions

We consider the following second-quantized Hamiltonian on the real line

$$H = \int_{\mathbb{R}} dx\, c^{\dagger}(x)\left(-\frac{\partial^2}{\partial x^2} + x\right)c(x), \tag{2}$$

where the Dirac fields obey the anticommutation relations $\{c(x), c^{\dagger}(y)\} = \delta(x - y)$, $\{c(x), c(y)\} = 0 = \{c^{\dagger}(x), c^{\dagger}(y)\}$. This model is free, i.e. quadratic in the fermions operators, and can be solved exactly. Indeed, introducing the modes

$$\psi^{\dagger}(\lambda) = \int_{\mathbb{R}} dx\, u(\lambda, x)c^{\dagger}(x) \qquad , \qquad \psi(\lambda) = \left(\psi^{\dagger}(\lambda)\right)^{\dagger}, \tag{3}$$

it is easy to show that $[H, \psi^{\dagger}(\lambda)] = \epsilon(\lambda)\psi^{\dagger}(\lambda)$, provided the single particle wave functions $u(\lambda, x)$ satisfy the Schrödinger equation

$$\left(-\frac{\partial^2}{\partial x^2} + x\right)u(\lambda, x) = \epsilon(\lambda)u(\lambda, x). \tag{4}$$

The solutions to this eigenvalue equation are well known to be Airy functions. Keeping only the eigenfunctions that decay to zero for $|x| \to \infty$:

$$u(\lambda, x) = \mathrm{Ai}(x + \lambda) \qquad , \qquad \mathrm{Ai}(x) = \int_{\mathbb{R}} \frac{dq}{2\pi} e^{i(qx + q^3/3)}. \tag{5}$$

Those solutions are parametrized by a real number $\lambda$, the eigenenergies are given by $\epsilon(\lambda) = -\lambda$. Hence the spectrum is continuous and unbounded. Due to the orthogonality relation $\int_{\mathbb{R}} dx\, u(\lambda, x)u(\mu, x) = \delta(\lambda - \mu)$, the modes satisfy the anticommutation relations $\{\psi(\lambda), \psi^{\dagger}(\mu)\} = \delta(\lambda - \mu)$. The Hamiltonian becomes diagonal in terms of the new modes

$$H = \int_{\mathbb{R}} d\lambda\, \epsilon(\lambda)\,\psi^{\dagger}(\lambda)\psi(\lambda) \qquad , \qquad \epsilon(\lambda) = -\lambda. \tag{6}$$

The ground state will play an important role in the following. It is a Dirac sea, obtained by filling all the states with negative energies (corresponding to $\lambda > 0$). The expectation values of the modes in this state are simply $\langle \psi^{\dagger}(\lambda)\psi(\mu)\rangle = \delta(\lambda - \mu)\Theta(\mu)$, where $\Theta$ is the Heaviside step function. The propagator is given by

$$G(x, x') \;=\; \langle c^{\dagger}(x)c(x')\rangle = \int_0^{\infty} d\lambda\, \mathrm{Ai}(x + \lambda)\mathrm{Ai}(x' + \lambda). \tag{7}$$

This is known as the Airy kernel [1]. Of course, for free fermions problems the two point function determines everything, more complicated observables reduce to determinants involving the propagator, by making use of Wick's theorem [12]. The operator $G_{\mathrm{Airy}}$ acting on functions in $L^2(\mathbb{R})$ as $G_{\mathrm{Airy}}f(x) = \int_{\mathbb{R}} dy\, G(x, y)f(y)$ can be seen as a filter, that projects the function $f(x)$ onto the subspace

$$\boxed{-\frac{d^2}{dx^2} + x \le 0.} \tag{8}$$

This simple observation will prove extremely useful in the following.

The kernel $G_{\mathrm{Airy}}$ admits several generalizations, which we now briefly discuss. The first one comes from introducing imaginary time operators $c^\dagger(x,\tau) = e^{\tau H}c^\dagger(x)e^{-\tau H}$, with propagator

$$G(x,\tau|x',\tau') = \langle c^\dagger(x,\tau)c(x',\tau')\rangle = \int_0^\infty d\lambda\, e^{-\lambda(\tau-\tau')}\mathrm{Ai}(x+\lambda)\mathrm{Ai}(x'+\lambda) \tag{9}$$

for $\tau' \le \tau$. This is known as the extended Airy kernel. The determinantal point process with correlation kernel $G(x,\tau|x',\tau')$ is called the Airy process [3]. It is also possible to look at finite temperature states, with averages taken as $\langle.\rangle_\beta = \mathrm{Tr}\left(.e^{-\beta H}\right)$, where the trace is taken over the underlying Fock space, and $\beta$ is the inverse temperature. In that case the mode occupation follows the Fermi-Dirac distribution, $\langle \psi^\dagger(\lambda)\psi(\mu)\rangle_\beta = \frac{\delta(\lambda-\mu)}{1+e^{-\beta\mu}}$, which leads to a generalization (see e.g. [13–15]) that interpolates between the Airy kernel (zero temperature, $\beta \to \infty$) and the Gumbel kernel (infinite temperature, $\beta \to 0$). In the following we stick to the Airy kernel (7), namely equal imaginary time and zero temperature.

## 1.2 Full counting statistics

Say we are interested in particle number fluctuations in an interval $A = [s,\infty)$ of $\mathbb{R}$. The natural object to consider is the following generating function

$$\Upsilon(\alpha,s) = \left\langle e^{\alpha\int_s^\infty dx\, Q(x)}\right\rangle \quad , \quad Q(x) = c^\dagger(x)c(x), \tag{10}$$

which is known as full counting statistics [16] in condensed matter literature. A standard computation gives

$$\Upsilon(\alpha,s) = 1 + \sum_{n=1}^\infty \frac{\alpha^n}{n!}\int_s^\infty dx_1\ldots\int_s^\infty dx_n\,\langle Q(x_1)\ldots Q(x_n)\rangle \tag{11}$$

$$= 1 + \sum_{n=1}^\infty \frac{(e^\alpha-1)^n}{n!}\int_s^\infty dx_1\ldots\int_s^\infty dx_n\,\langle : Q(x_1)\ldots Q(x_n):\rangle \tag{12}$$

$$= 1 + \sum_{n=1}^\infty \frac{(e^\alpha-1)^n}{n!}\int_s^\infty dx_1\ldots\int_s^\infty dx_n\,\det_{1\le i,j\le n} G(x_i,x_j) \tag{13}$$

$$= \det_s(I + (e^\alpha-1)G_{\mathrm{Airy}}). \tag{14}$$

Here $: Q(x_1)\ldots Q(x_n): = c^\dagger(x_1)\ldots c^\dagger(x_n)c(x_1)\ldots c(x_n)$ denotes normal ordering, (12) follows from (11) by applying Wick's theorem [12] and carefully rearranging the terms. In the last line, $\det_s$ is the Fredholm determinant of an integral operator acting on $L^2([s,\infty))$ with kernel $(e^\alpha-1)G(x,y)$. Eq. (13) can be taken as a definition of Eq. (14). The limit $E(s) = \lim_{\alpha\to-\infty}\Upsilon(\alpha,s)$ is the probability that the interval $A = [s,\infty)$ contains no fermions. Obviously $E(-\infty) = 0$ and $E(\infty) = 1$. This emptiness formation probability (EFP) is given by the exact formula

$$E(s) = \det_s(I - G_{\mathrm{Airy}}). \tag{15}$$

The EFP is the cumulative distribution of what is known as the GUE *Tracy-Widom distribution*, in particular it can be shown [1] to just equal (1). The corresponding probability density function (pdf), $p(s) = \frac{dE(s)}{ds}$, is illustrated in figure 1. It looks similar to a gaussian, but it is slightly skewed; in fact, one can show that $p(s) \sim e^{-(4/3)s^{3/2}}$ for $s \to \infty$ and $p(s) \sim e^{-(1/12)|s|^3}$ for $s \to -\infty$. It has mean $\langle s\rangle \simeq -1.771086$, variance $\kappa_2 = \langle(s-\langle s\rangle)^2\rangle \simeq 0.813194$, skewness $\kappa_3/(\kappa_2)^{3/2} \simeq 0.224084$ (positive means fatter tails on the right than on the left), and excess kurtosis $\kappa_4/(\kappa_2)^2 \simeq 0.093448$. Here $\kappa_3 = \langle(s-\langle s\rangle)^3\rangle$ and $\kappa_4 = \langle(s-\langle s\rangle)^4\rangle - 3(\kappa_2)^2$ are the third and fourth cumulants, respectively.

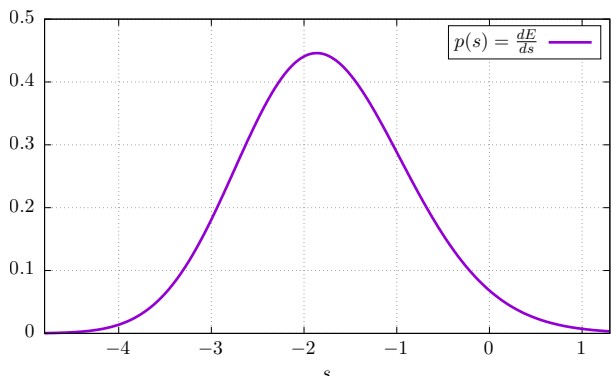

Figure 1: probability density function $p(s)$ of the GUE Tracy-Widom distribution.

It is not physically clear at this stage exactly of what this is a pdf in a realistic model. To clarify this point, we discuss now a simple example where T-W emerges.

### 1.3 Semiclassical analysis on a simple example

The Hamiltonian (2) looks utterly unphysical at first sight: the potential is linear, and does not even confine particles to a given region of space. Another related complication lies in the Dirac sea nature of the ground state, with infinite total particle number[1].

The case of a harmonic potential is better behaved, and also of unquestionable experimental relevance, through its relation to the Tonks-Girardeau gas (see [17] for a review). It turns out that the Airy Hamiltonian (2) describes the edge physics of the model in a harmonic potential, through a mechanism that we discuss below. To be more concrete, we now consider the Hamiltonian

$$H = \int_{\mathbb{R}} dx\, c^{\dagger}(x) \left( -\frac{\partial^2}{\partial x^2} + x^2 - \mu \right) c(x). \tag{16}$$

The parameter $\mu$ is a chemical potential, which allows to control the number of particles in the ground state. This model can be solved in a similar way as the previous one, and the single particle wave functions may be expressed in terms of Hermite polynomials. The problem is, in fact, formally identical to the well known quantum harmonic oscillator. Due to the confining nature of the potential, the energy levels are now discrete. Using this approach, one can for example show that the density of fermions in the ground state follows, when $\mu \to \infty$, the celebrated Wigner semi-circle law

$$\langle Q(x) \rangle = \frac{1}{\pi} \sqrt{\mu - x^2}. \tag{17}$$

**Bulk LDA—** It is enlightening to look at this problem using semiclassical analysis, sometimes also known as local density approximation (LDA) in cold atom literature. The key assumption is separation of scales: we look at mesoscopic scales around some point $x_0$, namely we look in an interval $[x_0 - \delta x, x_0 + \delta x]$, where $\delta x$ is much bigger that the mean distance between particles, and much smaller than the system size (both to be determined at this stage). On such distances the system looks homogeneous, with a well defined effective chemical potential $\mu_{\text{eff}}(x_0) = \mu - x_0^2$. The ground state propagator becomes the kernel of the projection onto

$$-\frac{d^2}{d(\delta x)^2} \leq \mu - x_0^2, \tag{18}$$

---

[1]One can show [11] that the particle number in the interval $[-a, \infty)$ diverges as $\frac{2}{3\pi} a^{3/2}$ for $a \to \infty$.

which is easy to determine. Indeed, thinking in Fourier space, the above becomes $k^2 < k_0^2$, where $k_0 = \sqrt{\mu - x_0^2}$, which defines a disk in phase space $(x, k)$. Hence the desired projector is given by

$$\langle c^\dagger(x_0 + \delta x)c(x_0 + \delta y)\rangle \quad = \quad \int_{-k_0}^{k_0} \frac{dk}{2\pi} e^{ik(\delta x - \delta y)} = \frac{\sin k_0(\delta x - \delta y)}{\pi(\delta x - \delta y)}, \qquad (19)$$

consistent with the claimed density (17). The particle number is then determined self consistently as $N = \int \langle Q(x)\rangle = \mu/2$, so the limit $\mu \to \infty$, where LDA is expected to become exact, is the thermodynamic limit $N \to \infty$ in the usual sense. The effective size of the system is then $L \sim (2N)^{1/2}$, while the mean interparticle distance is of order $a \sim N^{-1/2}$. The result (19) is therefore valid in the limit $N \to \infty$, $a \ll \delta x, \delta y \ll L$, and $k_0 > 0$.

**Edge from LDA—** The behavior close to the edge is slightly more complicated, but can still be obtained from semi-classical analysis (see Refs. [10, 18] for discussions). To explore this regime, we make the change of variable $x = \sqrt{2N} + \tilde{x}$, where the new variable is just assumed to be much smaller than system size for now. The propagator close to the edge becomes the kernel of the projection

$$-\frac{d^2}{d\tilde{x}^2} + \left(\sqrt{2N} + \tilde{x}\right)^2 \le 2N. \qquad (20)$$

Expanding the square, the term in $\tilde{x}^2$ is subleading compared to $2\sqrt{2N}\tilde{x}$, so may be discarded. After a final change of variable

$$\tilde{x} = \ell u \quad , \qquad \ell = (8N)^{-1/6}, \qquad (21)$$

the previous equation becomes

$$-\frac{d^2}{du^2} + u \le 0, \qquad (22)$$

whose kernel is precisely the Airy kernel studied in section 1.1, see (8). Back in $\tilde{x}$ coordinate system, this behavior occurs at scales of order $\ell = (8N)^{-1/6} \gg N^{-1/2}$, so does not contradict the bulk LDA argument, even though we are in a different regime with lower density now[2]. In this sense the limit is smooth, and LDA/semiclassics correctly predicts the edge behavior as a limit, since the result (22) can be proved by other means [1]. Semiclassically in phase space, we go from a disk $k^2 + x^2 \le \mu$ for the bulk to a parabolic region $q^2 + u \le 0$ for the edge, where $k, q$ are the momenta corresponding to $x, u$ respectively.

**The scale $\ell$—** It is important to realize that the above derivation does not rely on the fact that the potential be harmonic. For any reasonably behaved potential, we expect the same scaling behavior, since any smooth potential can be linearized close to the edge, and this will prove the dominant contribution. For this reason, Airy scaling close to the edge is expected to be quite generic. To the leading order, the only free parameter in such a mechanism is precisely the scale $\ell$ we calculated above in a particular example.

In terms of the bulk variables (recall $a$ is interparticle distance, and $L$ total system size) we have $\frac{\ell}{a} \propto \left(\frac{L}{a}\right)^{1/3}$, and for this reason the exponent $1/3$ is often seen as a hint to Airy scaling.

---

[2]In fact the edge limit can be seen as borderline with respect to separation of scales, since close to the edge interparticle distance is of order $\ell$, and density varies also on scales of order $\ell$.

**Tracy-Widom—** T-W appears when looking at the distribution of the rightmost particle. It may be determined by looking at the emptiness formation probability, which is given for finite $N$ by the Fredholm determinant $E_N(x) = \det_x(I - K_N)$, where $K_N$ is the kernel associated to the ground state propagator for the harmonic trap[3]. As we have just established, this kernel scales to the Airy kernel in a suitable edge limit, which means the (suitably rescaled) distribution of the rightmost fermion, $dE_N/dx$ converges to the Tracy-Widom distribution. Physically, the scale $\ell$ also controls the standard deviation of the distribution of the rightmost particle, which is given by $\sqrt{\langle x^2 \rangle_c} = a\ell + o(\ell)$, where $a \simeq 0.90177$, is the T-W standard deviation.

**Relation to GUE—** The free fermion problem looked at in the previous subsection is in fact formally identical to the random matrix problem where the T-W was originally discovered. Indeed, denote by $|\Psi\rangle$ the $N-$particle ground state of (16). In first quantization language, the many-body wave function reads $\phi(x_1, \ldots, x_N) = \langle c(x_1) \ldots c(x_N)|\Psi\rangle$ which is given by a Slater determinant. A direct calculation using properties of the Hermite polynomial and the Vandermonde identity shows

$$|\phi(x_1, \ldots, x_N)|^2 \propto \prod_{1 \le i < j \le N} (x_i - x_j)^2 e^{-\sum_{i=1}^N x_i^2}. \tag{23}$$

Therefore, the modulus square of the ground state wave function defines a joint pdf, which equals the joint pdf for GUE random matrices (the rhs) [19]. Using this observation, any statement for correlations of diagonal observables in the ground state may be turned into a random matrix theory problem, and vice versa. For example, the distribution of the rightmost particle becomes the distribution of the biggest eigenvalue in GUE language, which was exactly the problem originally studied by Tracy and Widom [1]. While this connection has been explored in several papers (see e.g. Refs. [20–23]), we do not need it here, and rely instead on standard quantum mechanics techniques to solve our quantum mechanics problems. From this perspective, Airy and T-W scaling follow at a fundamental level [10] from the free fermions Hamiltonian (2) and its Dirac sea ground state.

**Organization of the rest of the manuscript—** The remainder of the paper is devoted to the effect of interactions. We study in section 2 interacting models in traps at equilibrium, which can be seen as generalizations of (16), and demonstrate that T-W scaling generically survives at the edge (specific exceptions are discussed in appendix A). Section 3 tackles a more complicated quantum out of equilibrium problem, where the effects of interactions are subtle. In particular, we establish that the edge distribution has very long tail, in stark contrast with T-W. We conclude in section 4 and discuss some open problems.

## 2 Edge scaling for interacting quantum systems

As already mentioned an obvious question, left unanswered in the introduction, lies in the effect of interactions. We have discussed an explicit example, that has the Airy Hamiltonian (2) as effective edge Hamiltonian, but the free fermion structure was already built in, which means the distribution of the last particle could always be expressed as a Fredholm determinant. Showing convergence to the T-W distribution, ignoring mathematical difficulties, amounts to showing convergence of the propagator to the Airy kernel.

---

[3]Of course this propagator can be computed exactly for finite $N$ using orthogonal polynomial techniques, and this is by far the simplest way to prove convergence to T-W. This approach is however at odds with the nonrigorous LDA-based logic of the present paper, which will be the only game in town when considering interactions.

On the other hand, T-W is widely believed to be a universal distribution, and should also appear in problems where the free fermions structure is not already present in the microscopic model. For example, T-W scaling has been proved in the asymmetric exclusion processes (ASEP) for certain initial conditions, see e.g. [7]. The ASEP is related to the integrable XXZ spin chain, but away from the free fermion point. Despite these notable exceptions, there are in general still very few rigorous or exact results in this direction.

In the class of problems we look at, there is a simple argument explaining why T-W should appear at the edge of an interacting system (say) in a trap. Even if the underlying model may be extremely complicated, the edge is precisely the region where the density of particles becomes very low. In this region the quantum particles are diluted, and interactions with sufficiently fast decay, which might be very strong in the bulk, are expected to become weaker and weaker. Hence the particle become effectively free, and this makes generic T-W scaling behavior quite plausible. This mechanism is not much different from the usual appearance of a simpler effective field theory to describe the scaling limit of possibly extremely complicated microscopic models.

We discuss in this section an example where we are able to demonstrate this, and also, perhaps more importantly, are able to compute analytically the associated scale on which such behavior occurs. We do this using a combination of simple analytical arguments, backed by extensive numerical checks. Before doing that let us emphasize that the word *generic* in the previous paragraph is important. In fact, two clear exceptions will be discussed in subsections A.1 and A.2.

## 2.1 Lieb Liniger model in a harmonic trap

The first example we look at is the Lieb-Liniger model in a harmonic trap, governed by the second-quantized Hamiltonian[4]

$$H = \int dx \left( \Psi^\dagger(x) \left[ -\frac{\hbar^2}{2m} \frac{\partial^2}{\partial x^2} - \mu + V(x) \right] \Psi(x) + \frac{g}{2} \Psi^{\dagger 2}(x) \Psi^2(x) \right), \qquad (24)$$

with $g > 0$ (repulsive interactions). The field $\Psi$ is bosonic, it obeys the commutation relations $[\Psi(x), \Psi^\dagger(y)] = \delta(x - y)$, $[\Psi(x), \Psi(y)] = 0 = [\Psi^\dagger(x), \Psi^\dagger(y)]$. This model is well-known to be integrable in the absence of a trapping potential [24, 25]. The trap, however, typically breaks integrability. In the following, we will consider the (integrability breaking) harmonic trap which is the most natural and experimentally relevant.

Before proceeding any further, let us mention that this model is a natural generalization of the Fermi gas looked at in section 1.3, in the following sense. In the limit of infinitely strong repulsion, $g \to \infty$, the Tonks-Girardeau limit, the first quantized ground state bosonic wave function is given by [17]

$$\phi_b(x_1, \ldots, x_N) = \phi(x_1, \ldots, x_N) \prod_{1 \le i < j \le N} \text{sgn}(x_i - x_j), \qquad (25)$$

where $\phi$ denotes the fermionic wave function from section 1.3. For diagonal observables such as particle statistics, only the modulus square of (25) matters, so T-W describes the large $N$ edge behavior in the Tonks-Girardeau limit. For finite $g > 0$ the system is strongly interacting, and the wave function is more complicated.

**LDA and TBA—** Despite the fact that the system is not integrable, it is still possible to rely on separation of scales. As before, we assume that the system is sufficiently uniform on mesoscopic scales, which means it looks, locally, identical to the ground state of the Lieb-Liniger

---

[4]The first quantized form is $H = -\sum_{j=1}^{N} -\frac{\hbar^2}{2m} \frac{\partial}{\partial x_j^2} - \mu + V(x_j) + g \sum_{1 \le i < j \le N} \delta(x_i - x_j)$.

model without an external potential. This observation allows us to use the ground state thermodynamic properties of this Bethe-Ansatz integrable model. The thermodynamic Bethe Ansatz (TBA) description of homogeneous ground states is well known, see e.g. [25], and has been used to predict density profiles [26] and more complicated correlation functions [27,28] in the ground state.

In the following we work in units where $\hbar = m = 1$. For a given chemical potential $\mu$, the ground state is parametrized by a set of rapidities, that satisfy Bethe equations [24]. In the thermodynamic limit the relevant quantity is the density of rapidities $\rho(k, \mu)$, which can be shown to satisfy the linear integral equation (LIE)

$$\rho(k, \mu) - \frac{1}{2\pi} \int_{-k_F(\mu)}^{k_F(\mu)} K(k, q)\rho(q, \mu)dq = \frac{1}{2\pi}, \tag{26}$$

with kernel

$$K(k, q) = \frac{2g}{(k-q)^2 + g^2}. \tag{27}$$

Of great importance is also the energy of single particle excitations with quasimomentum $k$ above the ground state. It can be shown to satisfy another LIE

$$\varepsilon(k, \mu) - \frac{1}{2\pi} \int_{-k_F(\mu)}^{k_F(\mu)} K(k, q)\varepsilon(q, \mu)dq = k^2 - \mu. \tag{28}$$

$k_F(\mu)$ is defined through $\varepsilon(k_F, \mu) = 0$, for which also $\rho(|k| > k_F(\mu), \mu) = 0$. It plays a role similar to the Fermi momentum of free particles, and is therefore dubbed as such also in the presence of interactions. Since $k_F(0) = 0$, it also follows $k_F(\mu) \sim \sqrt{\mu}$ for small $\mu$, self-consistently from (28) (see Ref. [24] for a discussion).

The LDA assumption still allows to reconstruct the density profile by making the substitution $\mu \to \mu_{\text{eff}}(x) = \mu - x^2$ in the previous equations [26]. The main complication compared to section 1.3 is the dressing of thermodynamic quantities $\rho, \varepsilon$, through the kernel $K$, which means they cannot be obtained in explicit form in the bulk. The edge of the system is determined from $\mu_{\text{eff}}(x_e) = 0$, so it is located at $x_e = \pm\sqrt{\mu}$. The full density profile is given by

$$\langle Q(x) \rangle = \int_{-k_0(x)}^{k_0(x)} dk\rho(k, \mu_{\text{eff}}(x)) \quad , \quad k_0(x) = k_F(\mu_{\text{eff}}(x)) \underset{x \to \sqrt{\mu}}{\sim} \sqrt{\mu - x^2}, \tag{29}$$

with total particle number $N = \int_{-\sqrt{\mu}}^{\sqrt{\mu}} \langle Q(x) \rangle \, dx$.

**Edge scaling—** From the previous argument, we have determined that the edge is simply located at $x = \pm\sqrt{\mu}$, even though the full density profile can only be accessed in implicit form. The ground state is characterized in phase space by

$$\varepsilon(k, \mu_{\text{eff}}(x)) \leq 0 \quad , \quad \mu_{\text{eff}}(x) = \mu - x^2, \tag{30}$$

where $\varepsilon(k, \mu)$ is given by (28). Now comes the following simple but crucial point: at the edge the contribution from the integral in (28) vanishes to the leading order, so introducing $x = \tilde{x} + \sqrt{\mu}$ as before, we are left with the simple projection

$$k^2 + 2\sqrt{\mu}\tilde{x} \leq 0, \tag{31}$$

to the leading order in phase space. This is exactly the same result (8) as in the Tonks-Girardeau limit, hence interactions in the bulk, which are parametrized by $g/\sqrt{\mu}$, should not prevent the appearance of T-W scaling at the edge. We expect the subleading corrections due

to dressing to be no greater than those occuring already at the free fermion point. As before, the rightmost particle will be delocalized on scales of order $\ell = (4\mu)^{-1/6}$. Since the density profile close to the edge follows from the same argument, the T-W scaling is tightly related, from a more pedestrian perspective, to the behavior of the density close to the edge, which is the square-root scaling $\underset{x \to \pm\sqrt{\mu}}{\sim} \frac{1}{\pi}\sqrt{\mu - x^2}$, which turns out to be independent on interactions here. As a simple consequence, systems where the bulk density does not vanish as square-root are unlikely to yield T-W scaling.

It is also possible to interpret this result using field-theoretical language. An important property of interacting inhomogeneous systems in the Luttinger liquid universality class is that the Luttinger parameter, which parametrizes the strength of interactions, depends on position in the bulk [27, 28]. In such systems, the edge is precisely the place where it evaluates to one, the free fermion value (for inhomogeneous free fermions $K = 1$ throughout the system [29]). This argument should apply whenever the interaction between particles decays sufficiently fast. To illustrate this last point we discuss in appendix A.2 an example with inverse square long-range interactions, for which the Luttinger parameter can take other values at the edge.

## 2.2 XXZ spin chain in a slowly varying magnetic field

We study here another similar but more general example, this time of discrete nature. The Hamiltonian we consider is that of the spin-1/2 XXZ chain on the infinite lattice

$$H = \sum_{x \in \mathbb{Z} + 1/2} \left( S_x^{\mathrm{x}} S_{x+1}^{\mathrm{x}} + S_x^{\mathrm{y}} S_{x+1}^{\mathrm{y}} + \Delta(S_x^{\mathrm{z}} S_{x+1}^{\mathrm{z}} - 1/4) - h(x/R) S_x^{\mathrm{z}} \right), \tag{32}$$

where $S_j^\alpha = \frac{1}{2}\sigma_j^\alpha$, and $\sigma_j^\alpha$ act as Pauli matrices on the $j$'s copy of $\mathbb{C}^2$ and as identity on the others (we take the Hilbert space $(\mathbb{C}^2)^{\otimes L}$ and implicitly assume $L \to \infty$). Similar problems with spatially varying magnetic fields have been considered in the literature [29–31]. The magnetic field term depends on position, and plays a similar role as the trapping potential before. Before investigating this, let us summarize known results in the case of a constant magnetic field $h$. As is well known, the ground state has critical correlations for $|h| < 1 + \Delta$, well described by a Luttinger liquid field theory. For $|h| > 1 + \Delta$ the ground state is essentially fully polarized, all correlation functions are trivial. Recall also that $\Delta = 0$ can be mapped onto free fermions, while other values of $\Delta$ are interacting.

Now let us go back to a slowly varying magnetic field. We choose a continuous increasing function $h(u)$, that also, for later convenience, satisfies $h(-u) = h(u)$ and $\lim_{u \to \infty} h(u) = \infty$. The large parameter $R$ in (32) defines an effective system size, set by the location where $|h(x/R)| = 1 + \Delta$. Defining $x_{\mathrm{e}} = Rh^{-1}(1 + \Delta)$, inside the region $[-x_{\mathrm{e}}, x_{\mathrm{e}}]$ the system is inhomogeneous with critical correlations, outside it is a fully polarized product state.

**Bulk and edge TBA—** The TBA description of the ground state is also well known [25], and has a similar structure as the Lieb-Liniger one. It has also been checked numerically in Ref. [31], on the example $h(u) = u$, that the LDA approach gives the correct density profiles. With this at hand it is straightforward to look at the edge behavior, the calculations are exactly the same as in the previous subsection. With $x = x_{\mathrm{e}} + \tilde{x}$, we find the edge behavior in rapidity space

$$\frac{k^2}{2} + h'\left(\frac{x_{\mathrm{e}}}{R}\right)\frac{\tilde{x}}{R} \leq 0. \tag{33}$$

Assuming as before the emergence of Wick's theorem at the edge means we get T-W scaling. Introducing the new scale

$$\ell_\Delta = \left(\frac{R}{2h'(h^{-1}(1 + \Delta))}\right)^{1/3}, \tag{34}$$

and making the change of variables $\tilde{x} = \ell_\Delta u$, we recover the projector onto $-\frac{d^2}{du^2} + u \leq 0$. The scale $\ell_\Delta$ controls, as $\ell$ before, the standard deviation of the distribution of the last particle. It is now of order $R^{1/3}$, and depends explicitly on the interaction parameter $\Delta$. This prediction is tested numerically in the next subsection.

## 2.3 Numerical checks

The analytical argument presented in the previous subsection is quite heuristic. Indeed, we assumed free fermion behavior at the edge, and determined the propagator (correlation kernel) by using a self-consistent TBA description. This makes a numerical confirmation necessary.

Let us first note that numerical checks of Tracy-Widom scaling are notoriously difficult (see e.g. [32]). Since the associated scale is usually a power one third of the system size convergence is slow, even when reaching apparently very large system sizes. In classical setups Monte Carlo techniques are able to simulate large enough systems, however error bars tend to blur the results, especially when trying to extrapolate the data. The situation in the spin chain, we argue here, is slightly more favorable, which is one of the motivations for investigating T-W scaling in this quantum system. While the Hilbert space size naively grows exponentially fast, powerful variational techniques such as DMRG [33] are able to find the ground state with very good accuracy for large enough $R$. Efficient DMRG libraries able to implement continuous symmetries are now available in several programming languages (including Python [34] and C++ [35]), which simplifies our task considerably in the XXZ spin chain. The simulations shown below were performed using the C++ ITensor library [35].

For the magnetic field we made the choice $h(u) = u + au|u|$, which satisfies the hypothesis explained in the previous subsections. The term proportional to $u|u|$ might seem artificial, however, its presence ensures that the length scale (34) associated to T-W depends on $\Delta$ (for the linear potential $\ell_\Delta = (R/2)^{1/3}$ unfortunately does not depend on $\Delta$), and makes for a stronger numerical test of our analytical argument.

**Ground state density profile—** Let us first discuss the ground state magnetization profile $\langle S_x^z \rangle$, which is shown in figure 2 for several values of $\Delta$. The case $\Delta < -1$ leads to a trivial domain wall ground state, so we focus here on $\Delta > -1$. With our choice of magnetic field, an explicit computation solving a quadratic equation shows that the edge is located at

$$x_e = \pm R \frac{\sqrt{1 + 4a(1 + \Delta)} - 1}{2a}, \tag{35}$$

a prediction in very good agreement with numerics (note again the density profile for $a = 0$ has already been checked in Ref. [31]). The whole profile is also invariant under reflection symmetry $x \to -x$ conjugated with up-down (particle-hole after a Jordan-Wigner transformation) symmetry, due to the antisymmetry of the magnetic field we chose in (32). Such profiles are also related to equilibrium shapes of crystals in 2d, and have been investigated much earlier in this context [36].

We note in passing that another region develops in the middle of the chain for $\Delta > 1$, which has not been investigated to our knowledge in the spin chain. This is due to the fact that for $\Delta > 1 + h$, the homogeneous ground state is gapped with antiferromagnetic order. For $|h| > \Delta - 1$ this order is destroyed and we are back to the gapless phase. The interface defines in principle a new edge, which we do not discuss here. Let us just mention that such edge behavior is much more cumbersome to invertigate, and refer to [37] for a study in the (two periodic) classical dimer model where a similar phenomenon occurs. The edge defined by $x_e$ in (35) is not affected by this phenonenon, however, and this is what we focus on in the following.

**Sci**|**Post**                                                    SciPost Phys. 6, 057 (2019)

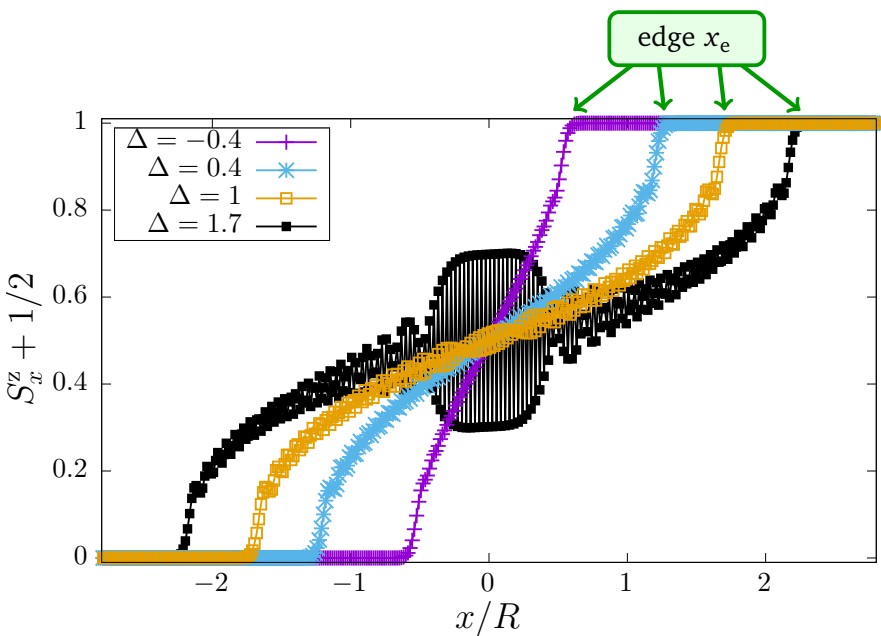

Figure 2: Numerical density profile for $R = 64$, and $a = 0.1$, obtained using DMRG. The data is shown for four different values of $\Delta$. In practice we use a system of total size $L = 512$, which is significantly larger than the effective size of the system $2x_{\mathrm{e}}$, outside of which the wave function is fully polarized. The central region with antiferromagnetic order is a specificity of $\Delta > 1$, as mentionned in the text. In the following we are interested in the behavior at the edge $x_{\mathrm{e}}$, indicated with green arrows.

**Edge distribution—** We now come to the actual check of our conjecture, which predicts T-W scaling with associated scale

$$\ell_\Delta = \left( \frac{R}{2\sqrt{1 + 4a(1 + \Delta)}} \right)^{1/3}. \tag{36}$$

Accessing the edge distribution can be done in a straightforward way in DMRG. We study the discrete analog of the "emptiness" formation probability, the probability that all spins at position $j \geq x$ be up,

$$E_x = \left\langle \prod_{j=x}^{\infty} \left( \frac{1 + 2S_j^z}{2} \right) \right\rangle, \tag{37}$$

close the right edge[5], see figure 2. The discrete probability density function (dpdf) is then reconstructed as $p_x = E_{x+1} - E_x$, and expected to converge to Tracy-Widom after proper rescaling involving $\ell_\Delta$. This is shown in figure. 3 (left). As can be seen the agreement is excellent. Note however a slight shift along the horizontal axis. We interpret this as a subleading order one additive correction to the $\ell_\Delta$ scaling, and checked that this is indeed a finite-size effect (not shown). The variance predicted by our analytical argument is also clearly confirmed by a finite-size scaling analysis, shown in figure. 3 (right). In this figure as well as in later plots, the leading correction is expected to be of order $R^{-2/3}$, and corresponds to the terms in $\tilde{x}^2$ that were discarded around Eq. (20) or (33).

---

[5]The left edge analog would be the probability $\left\langle \prod_{j=-\infty}^{x} \left( \frac{1-2S_j^z}{2} \right) \right\rangle$ that all spins are down, and leads to the same results by symmetry.

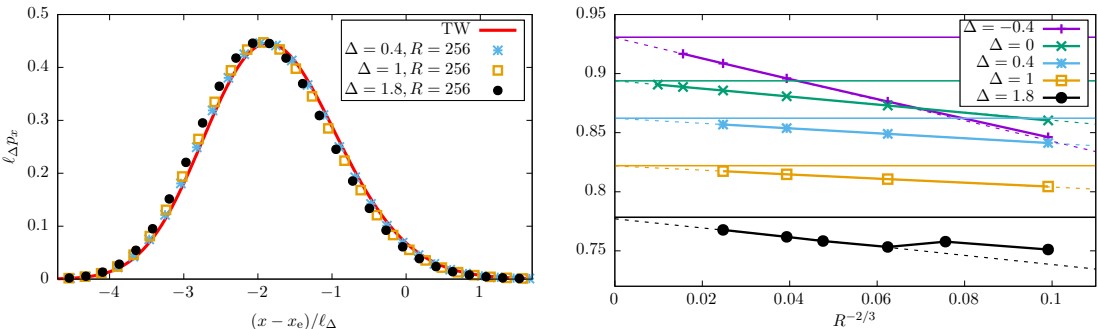

Figure 3: Left: rescaled dpdf for $R = 256$ and comparison with T-W (red line). Right: variance for $a = 1/10$ divided by the variance for $a = 0$, plotted as a function of $R^{-2/3}$ for several values of $\Delta$. The analytical prediction, given by $(1 + 4a(1 + \Delta))^{-1/3}$ is shown for comparison in thick horizontal lines. The data is extrapolated by a straight line, shown in dashed as a guide to the eye, with perfect agreement. The total chain length used for all simulations is $L = 8R$.

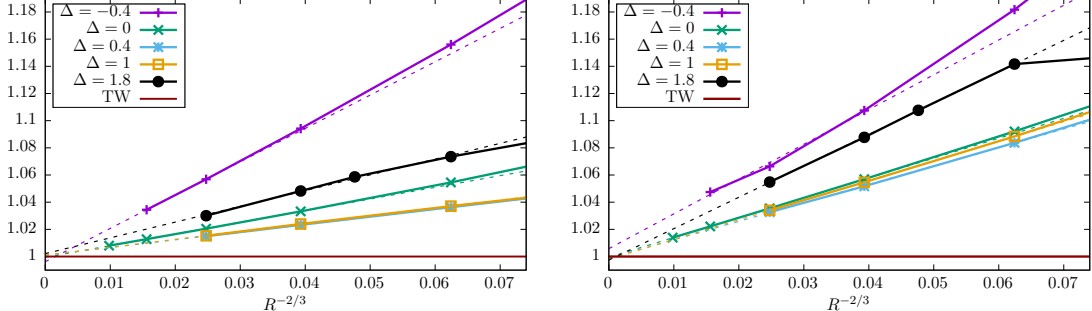

Figure 4: Left: skewness divided by the T-W one ($\simeq 0.224084$). Right: excess kurtosis divided by the T-W one ($\simeq 0.093448$). Both are shown as a function of $R^{-2/3}$. As in the previous figure, extrapolation to a straight line shows nearly perfect agreement.

To study more quantitatively the convergence to T-W, we also performed a finite-size scaling analysis of the skewness and excess kurtosis, related to the third and fourth cumulants (for a gaussian all cumulants of order larger than two are zero). This is shown in figure 4, with very convincing agreement. Relative errors for the largest system sizes we could access are typically 5% or less, depending on the value of $\Delta$. After extrapolation this error falls well under a percent in all cases, which is remarkable given the numerical difficulties usually associated to testing T-W. Of course, it is also possible to check higher order cumulants. However, those probe finer and finer details of the distribution, which would not be visible to the eye e.g. in figure 3. Since excess kurtosis shows larger errors than skewness, it is reasonable to expect finite-size effects to increase for higher order cumulants.

Let us finally mention that it is possible to use the spin chain to simulate the Lieb-Liniger model. This is done by considering the potential $h(u) = u^2$, and taking an appropriate low density limit (see [38, 39]). In that case simulations are typically limited to less than a hundred particles, we also checked that for reasonable interactions strength the skewness is within $\leq 10\%$ of T-W, with agreement improving for larger particle numbers.

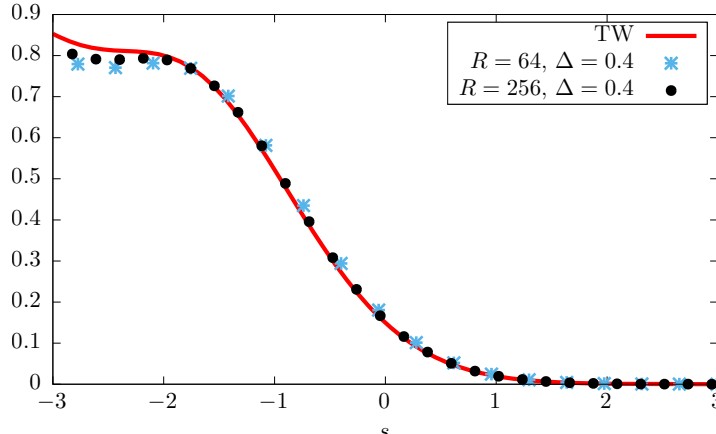

Figure 5: Rescaled entanglement entropy close to the right edge, which is expected to converge to (38) in the limit $R \to \infty$.

## 2.4 Universal entanglement profiles

We have argued in previous sections that interactions renormalize to zero close to the edge. This implies the emergence of the fermionic Wick theorem, a key ingredient to get T-W scaling. It is possible to check the fermionic Wick factorization property more explicitly, for example by looking at the entanglement entropy $S(x)$ of an interval $[x, \infty)$ for $x$ close to the (right) edge. For generic interacting systems computing this exactly is extremely complicated, however, for free (Airy or not) fermions it may be simply determined from the propagator [8,40,41], which leads us to conjecture that the formula

$$S\left(x_e + \ell_\Delta s\right) = -\text{Tr}_s \left[ G_{\text{Airy}} \log G_{\text{Airy}} + (I - G_{\text{Airy}}) \log(I - G_{\text{Airy}}) \right], \qquad (38)$$

holds for any $\Delta > -1$ in the limit $R \to \infty$ (the free case $\Delta = 0$ was previously established in Ref. [42]). Here $\text{Tr}_s$ denotes trace on $L^2[s, \infty)$. Note once again that only the scale $\ell_\Delta$ enters in the final result. Data for the rescaled entropy is shown in figure. 5. As can be seen the agreement is excellent and improves as $R$ is increased. We observe slight deviations when $s$ becomes large negative. This is expected since the entropy still sees bulk effects for finite-size. We note that the bulk entanglement entropy is more complicated in inhomogeneous systems, with even the free case turning out to be nontrivial [43]. For interacting systems the fact that the Luttinger parameter depends on position makes a field-theoretic treatment more difficult (see [28] for a discussion for local operators).

# 3 A Quantum out of equilibrium problem

We investigate in this section a different but related out-of-equilibrium setup, which shows interesting edge behavior. We consider the infinite XXZ spin-1/2 model (32) in the absence of a magnetic field. The system is initially prepared in the domain-wall state

$$|\Psi_0\rangle = |\dots \uparrow\uparrow\uparrow\uparrow\downarrow\downarrow\downarrow\downarrow \dots\rangle, \qquad (39)$$

and let evolve unitarily with the aforementioned Hamiltonian $H$ (the wave function at time $t$ is given by $|\Psi(t)\rangle = e^{-iHt} |\Psi_0\rangle$). At long times, a non trivial magnetization profile develops, with dynamical edges that we wish to study. As we shall see, away from free fermions ($\Delta = 0$) this will provide an example of a new universality class, beyond what is presently known. Before

entering into specifics, let us remind once again that T-W is not the only known universality class even in equilibrium problems, even though it is probably the most frequent/natural. To illustrate this, we discuss in appendix A two known exceptions to the scenario put forward in the previous section. The example discussed here will be of different nature, however.

The section is organized as follows. Several works have studied the spread of correlations after this quantum quench [44–55], we summarize the aspects that we need in section 3.1. Section 3.2 deals with previous results and claims for the edge behavior. We come in section 3.3 to our new results. In particular, we numerically access the real-space distribution of the rightmost up spin, the exact analog of what gave T-W in the previous section, or gives T-W for free fermions here. We show that this distribution is very delocalized, compared to other classes, and discuss in depth some of its properties. Finally, we summarize our findings in section 3.4.

## 3.1 Hydrodynamics and density profile

Despite the integrability of the XXZ chain and apparent simplicity of the quench protocol, exact computations of simple observables at finite time are extremely challenging, with only the return probability known in closed form [53]. A (generalized) hydrodynamic (GHD) description, able to tackle general such protocols, was put forward in Ref. [56, 57]. This approach is expected to become exact for our quench in the limit $x \to \infty, t \to \infty, x/t$ fixed, provided $|\Delta| < 1$, which we will assume from now on. It was used in Ref. [52] to compute the density profile analytically in that limit.

For the convenience of the reader, numerical examples of such density profiles are shown in figure 6 for several values of $\Delta$, and compared to the exact solution. The DMRG time evolution is implemented using the method of [58] together with the higher order Trotter formulas of [59]. The GHD limit for this quench is quite peculiar, and the density profile in the bulk region turns out to be nowhere continuous as a function of $\Delta$. This surprising behavior, reminiscent of Drude weight results [52, 60–66], which are believed to have also this property, is ultimately related to the quantum group structure [67] underlying the XXZ chain at root of unity.

We name the position $x_e$ where the GHD density profile vanishes the *GHD edge*. It is given by the simple formula $x_e/t = \sqrt{1 - \Delta^2}$ [52, 53]. There can be subleading corrections to this behavior. In fact, closer inspection of figure 6 (see in particular the inset) shows that density decays slowly for $x > x_e$ before it hits another edge at $x_f = t$ [48]. For $x > x_f$ the decay of the density appears to be super-exponential. Since the speed corresponding to $x_f$ can be interpreted as the group velocity $v_f = 1$ of a single magnon in a ferromagnetic background and does not depend on interactions, we dub $x_f = t$ the *free edge*. It can also be seen as a Lieb-Robinson-type bound in such a system. The fact that the GHD and free edge do not coincide away from $\Delta = 0$ will play an important role in the following.

## 3.2 Edge behavior of the density profile

T-W scaling for the edge front was established [8] at the free fermion point $\Delta = 0$ by an exact computation. However, such a scaling does not survive at the edge for $\Delta \neq 0$, as was argued in Ref. [52], the simplest reason being the fact that the density profile is linear at the GHD edge, not square-root as in all the examples discussed in the present paper, see e.g. (17). Such a linear behavior was also observed numerically in more complicated out-of-equilibrium setups [31]. An associated toy-model kernel [52], expected to qualitatively describe the GHD edge, was obtained from the exact computation of the density and current profiles. The calculation of those was formally identical to a different free fermion problem studied in Ref. [48, 49],

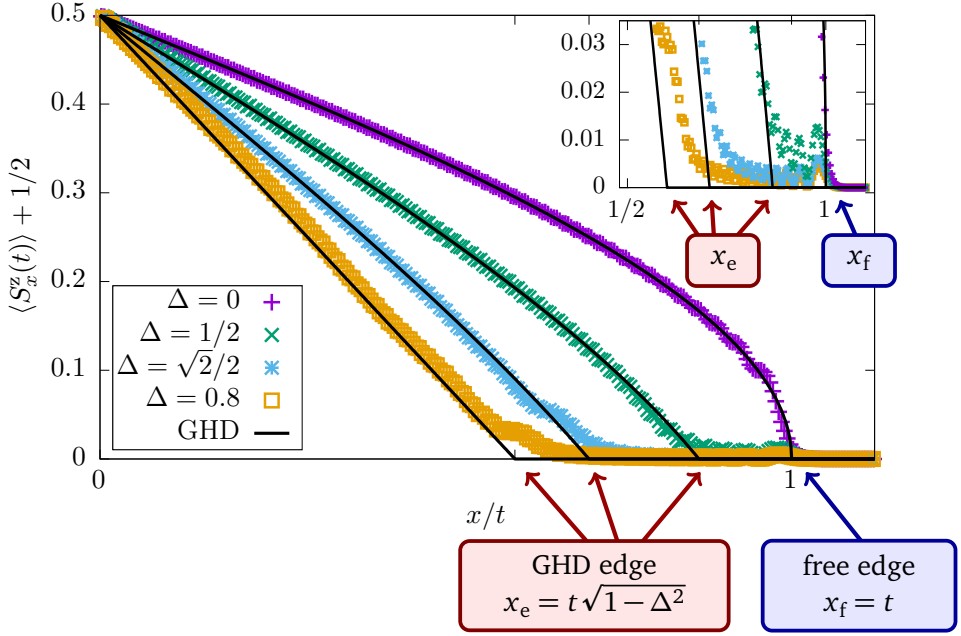

Figure 6: Numerical density profiles for $x > 0$, shown as a function of $x/t$ at time $t = 240$. The data is compared to the GHD formula $\langle S_x(t) \rangle = -\frac{q}{2\pi} \arcsin\left( \frac{\sin \frac{\pi}{q}}{\sin \gamma} \frac{x}{t} \right)$ for $\gamma/\pi = p/q$ irrational, $-\frac{x}{2t \sin \gamma}$ otherwise. Inset: zoom on the region $[x_e, x_f] = [t\sqrt{1-\Delta^2}, t]$ between the GHD edge and free edges, where the GHD prediction vanishes but subleading corrections remain. For $x > t$, density vanishes super-exponentially fast for all values of $\Delta$.

with time-dependent propagator

$$C(x, y|t) = \int_{-\pi}^{\pi} \frac{dk}{2\pi} \int_{-\pi}^{\pi} \frac{dq}{2\pi} e^{it(\cos k - \cos q) + ixk - iyq} f(k, q), \tag{40}$$

where

$$f(k, q) = \frac{\chi_\gamma(k, q)}{1 - e^{i(k - q + i0^+)}} + \text{reg}(k, q). \tag{41}$$

Here $\cos \gamma = \Delta$, and $\chi_\gamma(k, q) = 1$ if $|k|, |q| \in \{0, \gamma\} \cup \{\pi - \gamma, \pi\}$ and zero otherwise. $\text{reg}(k, q)$ denotes a function that is regular at $k = q$, but can have pointwise singularities, see [49] for explicit expressions. The asymptotics are then studied using standard saddle point techniques, where the singular term dominates. The case $\gamma = \pi/2$ yields exactly the $\Delta = 0$ domain wall quench, and the Airy kernel at the edge $x_e = t$ is derived by cubic expansion around $k, q = \pi/2$ in the phase (40). For $\gamma \neq \pi/2$ this point leaves the integration domain, and a quadratic expansion around $k, q = \gamma$ yields $C(x, y|t) = \frac{\pi}{\gamma \sqrt{t \cos \gamma}} \mathcal{E}\left( \frac{x - t \sin \gamma}{\sqrt{t \cos \gamma}}, \frac{y - t \sin \gamma}{\sqrt{t \cos \gamma}} \right)$, where $\mathcal{E}(X, Y)$ is the imaginary error kernel [52]

$$\mathcal{E}(X, Y) = \int_0^\infty d\lambda\, E(X + \lambda) E^*(Y + \lambda) \quad , \quad E(X) = \int_0^\infty \frac{dQ}{2\pi} e^{iQX + iQ^2/2}. \tag{42}$$

The scaling behavior close to the edge is $t^{1/2}$, instead of $t^{1/3}$. In our language, this can be naturally interpreted as the kernel of the projection $-i\frac{d}{dX} + X \leq 0$, consistent with the linear behavior for the density profile and the edge free fermions assumption. This analytical result in the toy-model is compared to numerical simulations in figure 7. As can be seen the agreement

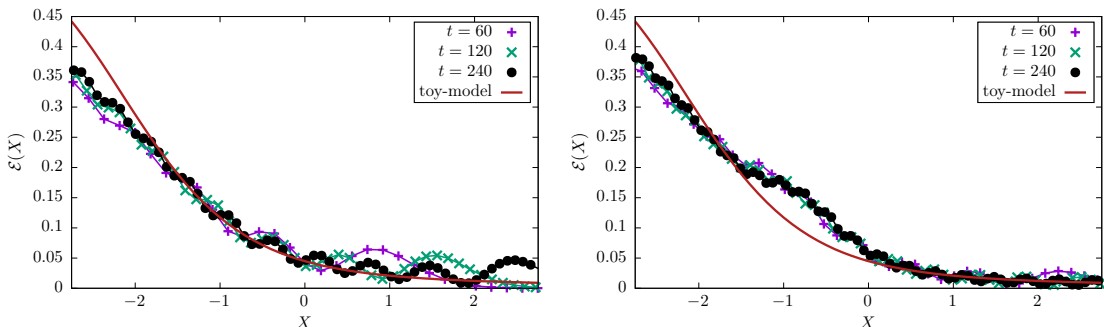

Figure 7: Rescaled density profiles around the GHD edge $x_e = t\sqrt{1-\Delta^2}$ for increasing times $t = 60, 120, 240$, and comparison with the kernel (42), shown in thick dark red. Left: $\Delta = 1/2$ shows good agreement, which improves for larger times. Right: $\Delta = 1/\sqrt{2}$. In that case the agreement is only fair, the difference with the free fermion kernel does not seem to go away in the limit $t \to \infty$.

is decent for $\Delta = 1/2$, but gets worse for larger values of $\Delta$, which means it is probably not exact. The collapse as a function of $\sqrt{t}$ seems quite good, however, sufficient to confidently exclude $t^{1/3}$.

What about the free edge, around which density is small but non-zero [48]? It was recently studied numerically in [55], where $t^{1/3}$ scaling close to $x_f$ was observed. The fact that a small fraction of quasiparticles go faster than the TBA/GHD speed was interpreted as a consequence of a slight order one excess in energy, due to the fact that $\langle \Psi_0 | H | \Psi_0 \rangle = -1/2$, where GHD implicitly assumes $\langle \Psi_0 | H | \Psi_0 \rangle = 0$.

We want to stress here that the observations made in Refs. [52, 55] are not incompatible, provided the results of [55] are interpreted carefully. First, the $t^{1/3}$ scaling is, in fact, also present in the free fermion propagator (40). Indeed, the result (42) was obtained from (41) by neglecting the regular terms, which provide subleading contributions. However, in the region $\frac{x_e}{t} < x < \frac{x_f}{t}$ this is not true anymore, since the indicator function $\chi_\gamma$ in (41) vanishes in the corresponding region of phase space. Close to $x = x_f = t$ the dominating saddle point is located at $k, q = \pi/2$, and yields a (subleading) product of two Airy functions, but not the Airy kernel. For $x/t > 1$, all correlations decay super-exponentially fast to zero.

From the previous considerations, it is not clear how the distribution of the rightmost particle would exactly look like, except for the fact that it should differ from T-W. This is the purpose of the next subsection, where we study it numerically for the first time, and point out an important analytical subtlety.

### 3.3 Distribution of the last particle

As our previous analysis suggests, the $t^{1/3}$ contribution close to the second edge should only account for a small fraction of *one* real-space particle, since it is subleading compared to the Airy kernel (recall T-W accounts for exactly one particle). This means the distribution of the last particle, the true analog of the Tracy-Widom distribution in our quench, should still be dominated by other effects, including diffusive effects in the neighborhood of the GHD edge $x_e$. This can be checked by once again computing the EFP, and numerically reconstructing the corresponding dpdf. The results are presented in figure 8 and show that the distribution is peaked around $x_e$. The free edge $x_f$ is then simply the termination of the right tail of the distribution.

While the collapse as a function of $\sqrt{t}$ near the GHD edge seems fair, it is unlikely that this fully describes the distribution of the last particle, due to the following analytical argument.

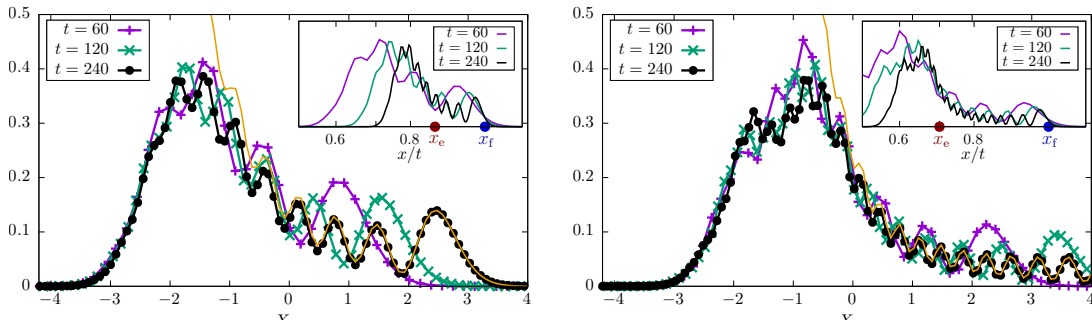

Figure 8: Rescaled distribution of the rightmost particle (rightmost up spin). As before the abscissa is $X = \frac{x - t \sin \gamma}{\sqrt{t \cos \gamma}}$, and data is shown for $\Delta = 1/2$ (left) and $\Delta = 1/\sqrt{2}$ (right). In both cases the collapse is good, and shows that the rightmost particle is mostly concentrated on the left of $x_\mathrm{e} = t \sin \gamma$ for the times we could access, even though its long right tail goes all the way to $x_\mathrm{free} = t$. The rescaled density profile for $t = 240$ is also shown in orange solid line for comparison (it is appropriately normalized to allow for comparison with the distribution). To help visualize the location of both edges (red $x_\mathrm{e}$ and blue $x_\mathrm{f}$ bullets), the same distribution is shown in the inset as a function of $x/t$.

Discarding the fact that the toy-model kernel (42) is unlikely to be exact for our quench, we find that it behaves for large $X, Y$ as

$$\mathcal{E}(X, Y) \quad \sim \quad \frac{\log X - \log Y}{4\pi^2 (X - Y)} \qquad , \qquad X \neq Y, \tag{43}$$

$$\mathcal{E}(X, X) \quad \sim \quad \frac{1}{4\pi^2 X}, \tag{44}$$

which is, importantly, *not integrable* for $X \to \infty$. This problem has to be cured by hand, introducing a hard cutoff at $x = t$ to make the density profile consistent with Lieb-Robinson bounds, but this would still mean that the figure above does not represent a true scaling function for the rescaled pdf. This suggests the possibility for logarithmic corrections in figure 8, which are hard to prove or disprove numerically.

These corrections should affect transport properties also; for example the particle number $N_\mathrm{dil}$ in the diluted region $[x_\mathrm{e}, \infty)$ was claimed to be of order one in Ref. [53], but if the true kernel decays as inverse distance as $\mathcal{E}$ does, then particle number should diverge logarithmically with time. As shown in figure. 9, this looks plausible numerically, back in the interacting quench.

Pushing the numerics further than done here is unfortunately unlikely to pay huge dividends. Indeed, we observe that convergence is quite slow in general, worse than regular T-W scaling encountered in this paper. In addition to the effects already mentionned, there are other competing terms, that are already present in the (probably simplified) free fermion model (40). In fact, we also checked that numerical convergence to the kernel (42) is already very slow in a discrete free fermion system modeling (42), even considering the very large times ($t > 1000$) we were able to access in that case.

## 3.4 Summary of our findings

Let us summarize our main numerical observations for $\Delta \neq 0$. For most values of $|\Delta| < 1$, and all accessible times most of the probability distribution is concentrated near the GHD edge $x_\mathrm{e} = t \sqrt{1 - \Delta^2}$. The distribution has an extremely long right tail, however, which extends all

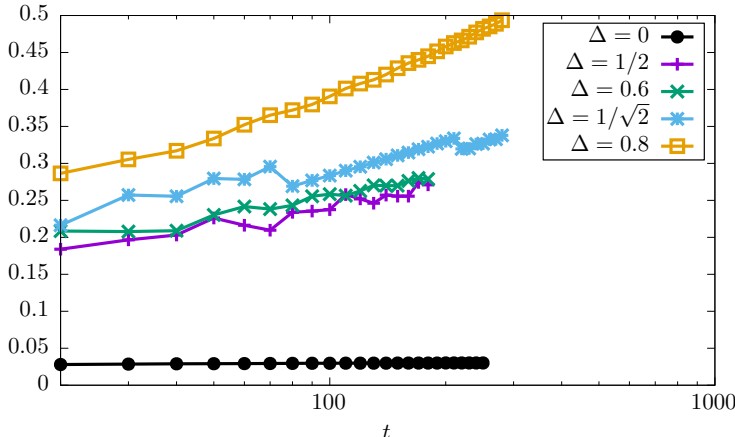

Figure 9: Particle number $N_{\mathrm{dil}}$ in the diluted region $[x_{\mathrm{e}}, \infty)$ on the right of the GHD edge, as a function of time (on a logarithmic scale). Data is shown for several values of $\Delta$. The results are consistent with a slow logarithmic divergence of this particle number, except at the free fermion point, $\Delta = 0$, where it saturates very quickly.

the way to $x_{\mathrm{f}} = t$. In stark contrast, the free fermion T-W distribution is concentrated on a much smaller region of width $t^{1/3}$ near the free edge (which coincides with the GHD edge, since $\Delta = 0$ in that case).

Motivated by the toy-model kernel of Eq. (42), which predicts a total particle number $N_{\mathrm{dil}} \propto \int_{x_{\mathrm{e}}+\epsilon}^{x_{\mathrm{f}}} \frac{dx}{x-x_{\mathrm{e}}} \propto \log(t/\epsilon)$ in the diluted region, we have observed numerically that particle number in the diluted region $[x_{\mathrm{e}}, \infty)$ grows with time, with a behavior consistent with a logarithmic divergence. This does suggest that the distributions shown in figure 8 might be far from converged, and might look different when the particle number becomes greater than one (for $|\Delta| \leq 0.8$ this should not happen before times of the order $t = 10^5$, a time which quickly increases as $\Delta$ is decreased). We expect the distribution to shift to the right, possibly even move away from the GHD edge at extremely large times. Since $\int_{x'_{\mathrm{e}}}^{x_{\mathrm{f}}} \frac{dx}{x-x_{\mathrm{e}}}$ does not diverge provided $\sqrt{1-\Delta^2} < x'_{\mathrm{e}}/t < 1$, we still expect the distribution to be at least supported on an interval $[x'_{\mathrm{e}}, x_{\mathrm{f}}]$, with an extremely long right tail. In all cases the free edge will correspond to the termination of the tail, which means the distribution is very different from T-W.

To further illustrate this last point, we have computed numerically the variance and skewness of the distribution, as a function of time. The results are shown in figure. 10. The variance grows possibly as fast as $t^2$ (or slightly slower), while skewness keeps on increasing: we find once again a behavior consistent with a logarithmic divergence, very different from the T-W finite value which is about 0.224084.

Given the many pitfalls described above, numerics alone are unlikely to give a definite answer; clearly better analytical insights are needed to explore those new classes of edge behavior – we describe possible strategies in the conclusion. Let us finally mention that unitary dynamics might be crucial to obtain such types of behavior. Indeed, the ancillary fermion model we relied on, seen as an equilibrium problem, is non analytic in Fourier space, which means it cannot be obtained as a ground state of an Hamiltonian with *local* interactions. For similar reasons, it is not completely clear whether the final answer for correlations at the GHD will be free fermionic or not. Correlations near the free edge should be, however.

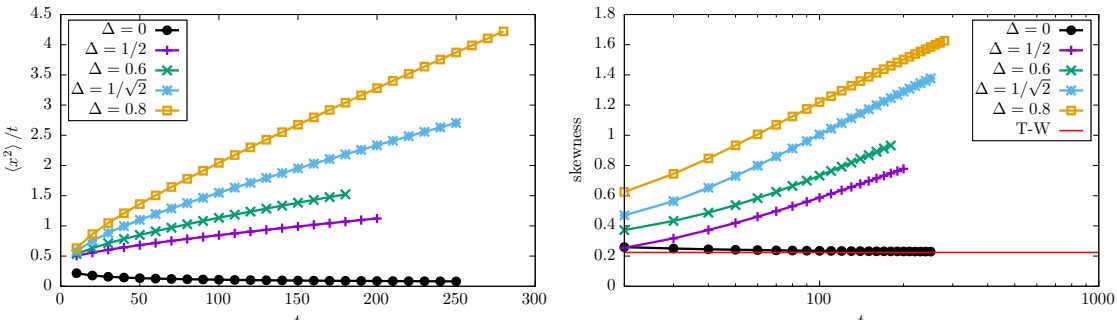

Figure 10: Left: variance divided by $t$, as a function of time $t$. As can be seen, the distribution is much more delocalized than expected from regular diffusion. The free fermion point is also shown for comparison, in that case we expect a decay as $t^{-1/3}$. Right: skewness as a function of time (on a logarithmic scale). It appears to grow slowly with time, possibly logarithmically. The skewness of T-W, which is much smaller, is also shown for comparison, and matches very well the free fermion calculation.

## 4  Conclusion

In this paper, we have investigated a few simple inhomogeneous interacting quantum systems in traps, and their edge properties. Our main result is extremely simple to formulate: at the edge the particle density goes to zero, so sufficiently local interactions are also renormalized to zero. While this observation is well-known from standard TBA arguments, the fact that it holds at a subleading scale is perhaps underappreciated. This partly explains the universality of such edge distributions, in particular T-W. In our case its appearance is ensured by the validity of the LDA (or semiclassical) hypothesis in the bulk, and then taking the edge limit, which is smooth. More importantly, the LDA/TBA approach also allowed us to compute exactly the length scale associated to T-W, essentially the only free parameter for such scaling. All those claims were carefully checked by large scale DMRG calculations in a spin chain model, that also admits Lieb-Liniger as a limit.

It is of course difficult to prove our semiclassical treatment, since the system is non integrable, but already a proof for discrete inhomogeneous spin chains that map to free fermions would be very interesting. Note also that the argument should carry over to inhomogeneous quantum systems whose homogeneous analogs are not integrable, but in that case we would not be able to compute analytically the location of the edge and the scale associated to T-W, as we did in the present paper.

There are several interesting directions for future investigations, let us mention some of those now. First, we only looked at ground states here, but it would be interesting to investigate finite-temperature effects, and see whether the Hamiltonian (2) still emerges at the edge in the presence of interactions. Even though the edge effects are too small to be accessible to current cold-atom experiments at small but finite temperature, such a result would nevertheless provide a clear experimental prediction.

A better understanding of edge universality classes in out-of-equilibrium quantum problems is obviously left as an important open problem. For the quench from a domain wall state the edge distribution can in principle be computed exactly using the method put forward in [53,68–70], and applying it to the exact EFP in the six vertex model with domain wall boundary conditions, for which multiple integral exact formulas are available [71,72]. This might provide a way to rigorously study those new edge universality classes for any value of $\Delta$, but technical difficulties, while we believe not insurmountable, remain formidable. A

more heuristic approach would be to better understand the corrections to GHD, which are less understood in the diluted regime we are interested in.

The point $\Delta = 1$ is also of great interest, especially given the fact that the (sub-ballistic) transport properties of this point are still theoretically not so well understood for the pure states [50, 51, 53, 54, 73] encountered here. Studying the distribution of the rightmost particle in that case would be of great interest; since the signal still spreads ballistically, we expect an even more spectacular long tail effect, possibly related to the difficulties in reliably extracting a (non-overestimated) transport exponent. The long tail effect should also be present for $|\Delta| > 1$, even though there is no transport in this quench.

Let us finally emphasize that we have looked here at pure states, that have zero entropy in string-TBA language. Finite entropy states are more relevant when the system is prepared in a thermal density matrix. This means there is no direct connection between our edge behaviors and the corrections to GHD studied in [74], or the transport studies [75, 76] at the Heisenberg point. Finally, investigating edge distributions in chaotic systems would be highly desirable, also in relation to operator spreading. Looking at those problems with the perspective of the present paper should shed some light on these timely issues.

**Acknowledgments.** I thank Filippo Colomo, Andrea De Luca, Jacopo De Nardis, Jérôme Dubail, Alexandre Lazarescu, Grégoire Misguich, Herbert Spohn, Eric Vernier and Jacopo Viti for enlightening discussions. I am also grateful to Jérôme Dubail and Herbert Spohn for a careful reading of the manuscript. The DMRG calculations were performed using the ITensor C++ library [35].

# A   Other universality classes

We have demonstrated in this paper how T-W naturally emerges at the edge of an inhomogeneous interacting system. Our main motivation was to partially fill a gap in the literature, and focus on interacting quantum system at equilibrium, which have not been much investigated in this context. This does not mean that T-W scaling is systematic, as we briefly discuss here, however. In appendix A.1 we look at simple free fermions problems that do not exhibit T-W behavior, but are described by higher order free fermions kernels. Appendix A.2 deals with the Calogero-Sutherland-Moser model, which belongs to the universality class of $\beta$-matrix ensembles, which is not free fermions. An even more spectacular and less understood exception is discussed in section 3 in the main text.

## A.1   Tuning the dispersion relation

Let us go back to the spin chain in a magnetic field studied in section 2.2. As is well known, the point $\Delta = 0$ can be mapped onto free fermions, upon performing a Jordan-Wigner transformation. In terms of lattice fermions $\{c_i, c_j^\dagger\} = \delta_{ij}$ the Hamiltonian reads

$$H = \frac{1}{2} \sum_x \left( c_{x+1}^\dagger c_x + c_x^\dagger c_{x+1} - h(x/R)(2c_x^\dagger c_x - 1) \right). \tag{45}$$

The homogeneous case (constant $h$) can be solved by going to Fourier space. The dispersion relation reads in that case $\varepsilon(k) = \cos k - h$. For a varying magnetic field, LDA tells us the ground state propagator is the kernel of the projection $\cos k - h(x/R) < 0$. Near the edge $x_e = \pm R h^{-1}(1)$, the cosine may be expanded to second order at $k = 0, \pi$, and we recover T-W scaling.

It is of course possible to consider different dispersion relations, which correspond to adding next nearest neighbors hoppings. For example the choice $\varepsilon(k) = \cos k - \frac{1}{4}\cos 2k$ is quartic around $k = 0$, $\epsilon(k) = 3/4 - k^4/8 + O(k^6)$. This means the corresponding edge behavior will be governed by the kernel of the projection

$$\frac{1}{8}\frac{d^4}{dx^4} + \frac{x}{R} \leq 0, \tag{46}$$

which implies $R^{1/5}$ scaling at the edge, instead of $R^{1/3}$. The distribution of the rightmost particle will then be given by a different distribution, built with a kernel constructed from functions $A_5(u) = \int_{\mathbb{R}}\frac{dq}{2\pi}e^{iqu+iq^5/5}$, instead of Airy functions. This kernel has been studied in a slightly different free fermions context in [77].

Several other examples have been found in statistical mechanical literature, in particular in relation to limit shapes. Those include the Pearcey kernel [78] for quartic singularities (Airy is cubic), or the tacnode kernel [79] (which includes, roughly speaking, quadratic band touching). We refer to [80] for a review of these free fermionic universality classes.

## A.2   Calogero-Sutherland models and $\beta$-matrix ensembles

Another exception to our previous discussion is provided by the Calogero-Sutherland-Moser model [81] in a harmonic trap, with first quantized Hamiltonian

$$H = \sum_{j=1}^{N}\left(-\frac{\partial^2}{\partial x_j^2} + x_j^2\right) + \sum_{i\neq j}\frac{\beta(\beta/2-1)}{(x_i-x_j)^2}. \tag{47}$$

This is a long range interacting system for $\beta \neq 2$, to which our previous renormalization argument does not apply. Contrary to models with short-range interactions such as Lieb-Liniger, the diluted particles close to the edge might still interact strongly with their bulk counterparts, so we do not necessarily expect free fermions factorization.

It can be shown analytically that this is precisely what happens. For inverse square interactions the wave function can be obtained exactly, and its modulus square given by

$$|\psi_{\text{cs}}(x_1,\ldots,x_N)|^2 \propto \prod_{1\leq i<j\leq N}(x_i-x_j)^\beta e^{-\sum_{j=1}^N x_j^2}. \tag{48}$$

The joint pdf on the rhs is known as $\beta$-ensemble in random matrix theory context. The corresponding distribution of the last particle satisfies a $\beta$-deformed Tracy-Widom distribution, see e.g. [82–84] ($\beta = 2$ is the T-W discussed in the present paper). Hence for $\beta \neq 2$ the edge behavior lies in a different universality class which is not free fermions anymore.

It is useful to interpret this result in terms of Luttinger parameter, which parametrizes the strength of interaction in field theoretical language. Due to the rather explicit nature of the wave function, correlation functions can be calculated exactly, and the Luttinger parameter extracted from the corresponding exponent. It turns out that, contrary to the cases studied before, the Luttinger parameter stays constant throughout the system, $K = \frac{2}{\beta}$. Presumably, an interaction with faster decay than inverse square would recover a Luttinger parameter that varies with position, and evaluates to $K = 1$, the free fermion value, at the edge. Checking this idea numerically seems quite difficult, however. Let us remark that it is not clear how one can obtain general $\beta$-T-W scaling with the type of condensed matter systems we look at in the present paper, except for the –clearly fine-tuned– example discussed here.

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
