# Peer review of "Free fermions at the edge of interacting systems"

_SciPost Physics, doi:SciPost Phys. 6, 057 (2019)_

## Round 2 · Referee Report · Anonymous (Referee 1) · 2019-2-21

Strengths

-

Weaknesses

-

Report

Tracy-Widom distribution is ubiquitous in the quantitative description
of the scaling behaviour at the edge of many (mainly free-fermionic)
one-dimensional quantum systems. A natural and still open question is
how much such universal behaviour is affected by the presence of
interactions.

Motivated by the simple heuristic idea that, since density vanishes at
the edge, to leading order, interactions that are present in the bulk
may become irrelevant there, in the first part of the paper the author
addresses the question by a detailed investigation of various
interacting one dimensional quantum models, such as Lieb-Liniger in a
trap, or XXZ quantum spin chain in a spatially varying magnetic field.
This is done by combining semiclassical and thermodynamic Bethe Ansatz
arguments, with careful numerical check. The whole analysis, although
non rigorous, brings additional strong and convincing support to the
idea that Tracy-Widom behaviour at the edge may survive even in the
presence of interactions (modulo some caveat, illustrated by two
examples in Appendix). Although not offering in my opinion much
exciting novelty, this first part of the paper puts the subject into a
nice perspective, and provides a precise definition of mathematical
ingredients, together with a clear and sound explanation of physical
ideas.

Next, the author adresses the question in a different but related
out-of-equilibrium setup, namely that of the quantum quench of an
infinite XXZ spin-1/2 quantum chain, in the absence of magnetic field,
initially prepared in a domain-wall state. The system is known to
develop a non trivial magnetization profile, with dynamical edges
(`light-cone'). As known from previous analytic works, in such setup
Tracy-Widom scaling is indeed observed at the edge in the
free-fermionic case $\Delta=0$, but it does not survive in the
interacting case $\Delta\not=0$. Thus the main heuristic idea at the
basis of the first part of the paper is no more valid in such
non-equilibrium setup. In this second part, the paper does not add
much analytic insights, but the numerical results presented are by far
the most refined and accurate I have ever seen on the subject, and
unveil, when $\Delta\not=0$, new intriguing features of the edge
density behaviour in the domain-wall quantum quench.

The paper is well written and I enjoyed the reading. I really
appreciated the concise but extremely clear and precise introduction
to the Tracy-Widom distribution.

In conclusion, this is accurate, well done, and nicely presented work,
which offers a unified view on an important open issue (robustness of
of Tracy-Widom edge behaviour in presence of interaction), and
provides new numerical results of great interest to the community,
that will surely stimulate further analytic and numerical
investigations.

Requested changes

I have noticed various typos, and a couple of somewhat unclear paragraphs,
but now I see these have all been corrected in version 2.

---

## Round 2 · Referee Report · Anonymous (Referee 2) · 2019-2-25

Strengths

1) Interesting topic, first systematic calculation of edge behaviour for interacting systems 2) Readable and self-consistent presentation

Weaknesses

Only minor issues, see below

Report

The author studies the edge behavior in inhomogeneous 1D interacting systems,
where the edge is defined by the location where the particle density vanishes.
The inhomogeneity is generated by a trapping potential for the Lieb-Linger
model, or a spatially varying magnetic field for the XXZ spin chain.
In the absence of interactions, the edge behavior is, in general, governed
by the Tracy-Widom scaling, which gives e.g. the probability distribution
of the last particle. The main argument of the paper is that, even in
the presence of interactions, the edge corresponds to a dilute regime of
quasiparticles, where the interactions effectively renormalize to zero,
hence the Tracy-Widom scaling should apply. This prediction is then tested
on the inhomogeneous XXZ chain, by carrying out DMRG calculations of the
emptiness formation probability (EFP). Here the only free parameter is the
length scale associated to the edge, which is nicely recovered in the numerics.

On the other hand, the case of a dynamical edge appears to be more
complicated. The simplest scenario studied here is the time evolution
from a domain-wall initial state in the XXZ chain. Indeed, the solution
via generalized hydrodynamics (GHD) predicts an edge location, which
does not really correspond to the location where the density vanishes.
Instead, a “free edge” seems to appear beyond the GHD edge, the location
of which corresponds to the free magnon velocity. Around the GHD edge,
the scaling appears to be diffusive, described by an imaginary error kernel.
The very long tail extending up to the free edge is interpreted as a subleading
effect, which might still show the $t^{1/3}$ length scale as observed recently.
Moreover, it is pointed out that the particle number beyond the GHD edge might
have a logarithmic divergence which seems to be compatible with the numerics.

I think this is a very interesting and well-written paper, which studies
a problem that has not yet been considered systematically before.
The manuscript clearly deserves to be published, the only minor
issues I found are listed below.

Requested changes

1) I guess in Eq. (31) the second argument of the integrand should read $\mu_{eff}(x)$ instead of $k_0(x)$.

2) In Eq. (33) the variable should be $\tilde x$ and not $x$, right? Regarding this equation, I was wondering what is the lowest order correction due to the dressing? Is it of the same order of magnitude (i.e. $\tilde x^2$), as the other correction neglected here?

3) In section 2.4 the entropy profiles are considered and the difference of edge and bulk behaviour is discussed. For free fermions, such a study was already carried out in J. Stat. Mech. (2014) P04005.

4) At the beginning of section 3.3 the formulation is a bit unclear. It is stated that the second edge is subleading compared to the Airy kernel. However, in this case one should rather compare to the kernel (44), since this is the one that is supposed to describe the GHD edge. Also, the comment that the Airy kernel provides one particle is not fully correct, since it actually describes many particles close to the edge, and not only the last one.

5) Some typos I discovered: - Above Fig. 3: “As can be seen the agreement in excellent” - Fig. 5 caption: “expected to to converge” - before (44): “is the the imaginary error kernel”

---

## Round 3 · Author Response

I am grateful to both referees for their reports, and their very positive appreciation of the manuscript. I essentially agree with all their comments.

I am happy to submit a new version, with various small improvements.

---

## Round 3 · List of Changes

I made the following changes in response to the remarks by the second referee.

1) and 2) This is correct, these typos are now corrected. Regarding the question, I believe the corrections are no greater than those for free fermions. 3) I am grateful to the referee for pointing out this reference, which I had missed. This work is now explicitly mentioned in the text. 4) I tried to improve this subsection, as well as several other related points in the manuscript. The discussion is hopefully easier to follow now. The referee is of course right that the Airy kernel accounts for an infinite number of particles, not one as previously stated. This is corrected now; what I meant to point out was that the T-W distribution itself accounts for only one particle, in references to other claims such as those of Ref. 55. 5) Those are now fixed.

I also made other minor changes not requested by the referees.

---

## Editorial Decision

published